# Quantized Random Projections and Non-Linear Estimation of Cosine Similarity

**Ping Li**
Rutgers University
pingli@stat.rutgers.edu

**Michael Mitzenmacher**
Harvard University
michaelm@eecs.harvard.edu

**Martin Slawski**
Rutgers University
martin.slawski@rutgers.edu

## Abstract

Random projections constitute a simple, yet effective technique for dimensionality reduction with applications in learning and search problems. In the present paper, we consider the problem of estimating cosine similarities when the projected data undergo scalar quantization to $b$ bits. We here argue that the maximum likelihood estimator (MLE) is a principled approach to deal with the non-linearity resulting from quantization, and subsequently study its computational and statistical properties. A specific focus is on the on the trade-off between bit depth and the number of projections given a fixed budget of bits for storage or transmission. Along the way, we also touch upon the existence of a qualitative counterpart to the Johnson-Lindenstrauss lemma in the presence of quantization.

## 1 Introduction

The method of random projections (RPs) is an important approach to linear dimensionality reduction [23]. RPs have established themselves as an alternative to principal components analysis which is computationally more demanding. Instead of determining an optimal low-dimensional subspace via a singular value decomposition, the data are projected on a subspace spanned by a set of directions picked at random (e.g. by sampling from the Gaussian distribution). Despite its simplicity, this approach comes with a theoretical guarantee: as asserted by the celebrated Johnson-Lindenstrauss (J-L) lemma [6, 12], $k = O(\log n/\varepsilon^2)$ random directions are enough to preserve the squared distances between all pairs from a data set of size $n$ up to a relative error of $\varepsilon$, irrespective of the dimension $d$ the data set resides in originally. Inner products are preserved similarly. As a consequence, procedures only requiring distances or inner products can be approximated in the lower-dimensional space, thereby achieving substantial reductions in terms of computation and storage, or mitigating the curse of dimensionality. The idea of RPs has thus been employed in linear learning [7, 19], fast matrix factorization [24], similarity search [1, 9], clustering [2, 5], statistical testing [18, 22], etc.

The idea of data compression by RPs has been extended to the case where the projected data are additionally quantized to $b$ bits so as to achieve further reductions in data storage and transmission. The extreme case of $b = 1$ is well-studied in the context of locality sensitive hashing [4]. More recently, $b$-bit quantized random projections for $b \geq 1$ have been considered from different perspectives. The paper [17] studies Hamming distance-based estimation of cosine similarity and linear classification when using a coding scheme that maps a real value to a binary vector of length $2^b$. It is demonstrated that for similarity estimation, taking $b > 1$ may yield improvements if the target similarity is high. The paper [10] is dedicated to J-L-type results for quantized RPs, considerably improving over an earlier result of the same flavor in [15]. The work [15] also discusses the trade-off between the number of projections $k$ and number of bits $b$ per projection under a given budget of bits as it also appears in the literature on quantized compressed sensing [11, 14].

In the present paper, all of these aspects and some more are studied for an approach that can be substantially more accurate for small $b$ (specifically, we focus on $1 \leq b \leq 6$) than those in [10, 17, 15]. In [10, 15] the non-linearity of quantization is ignored by treating the quantized data as if they had been observed directly. Such "linear" approach benefits from its simplicity, but it is geared towards fine quantization, whereas for small $b$ the bias resulting from quantization dominates. By contrast, the approach proposed herein makes full use of the knowledge about the quantizer. As in [17] we suppose that the original data set is contained in the unit sphere of $\mathbb{R}^d$, or at least that the Euclidean

norms of the data points are given. In this case, approximating distances boils down to estimating inner products (or cosine similarity) which can be done by maximum likelihood (ML) estimation based on the quantized data. Several questions of interest can be addressed by considering the Fisher information of the maximum likelihood estimator (MLE). With regard to the aforementioned trade-off between $k$ and $b$, it turns out that the choice $b = 1$ is optimal (in the sense of yielding maximum Fisher information) as long as the underlying similarity is smaller than $0.2$; as the latter increases, the more effective it becomes to increase $b$. By considering the rate of growth of the Fisher information near the maximum similarity of one, we discover a gap between the finite bit and infinite bit case with rates of $\Theta((1 - \rho_*)^{-3/2})$ and $\Theta((1 - \rho_*)^{-2})$, respectively, where $\rho_*$ denotes the target similarity. As an implication, an exact equivalent of the J-L lemma does not exist in the finite bit case.

The MLE under study does not have a closed form solution. We show that it is possible to approximate the MLE by a non-iterative scheme only requiring pre-computed look-up tables. Derivation of this scheme lets us draw connections to alternatives like the Hamming distance-based estimator in [17].

We present experimental results concerning applications of the proposed approach in nearest neighbor search and linear classification. In nearest neighbor search, we focus on the high similarity regime and confirm theoretical insights into the trade-off between $k$ and $b$. For linear classification, we observe empirically that intermediate values of $b$ can yield better trade-offs than single-bit quantization.

**Notation**. We let $[d] = \{1, \ldots, d\}$. $I(P)$ denotes the indicator function of expression $P$. For a function $f(\rho)$, we use $\dot{f}(\rho)$ and $\ddot{f}(\rho)$ for its first resp. second derivative. $\mathbf{P}_\rho$ and $\mathbf{E}_\rho$ denote probability/expectation w.r.t. a zero mean, unit variance bivariate normal distribution with correlation $\rho$.

**Supplement**: Proofs and additional experimental results can be found in the supplement.

## 2 Quantized random projections, properties of the MLE, and implications

We start by formally introducing the setup, the problem and the approach that is taken before discussing properties of the MLE in this specific case, along with important implications.

**Setup.** Let $\mathcal{X} = \{x_1, \ldots, x_n\} \subset \mathbb{S}^{d-1}$, where $\mathbb{S}^{d-1} := \{x \in \mathbb{R}^d : \|x\|_2 = 1\}$ denotes the unit sphere in $\mathbb{R}^d$, be a set of data points. We think of $d$ being large. As discussed below, the requirement of having all data points normalized to unit norm is not necessary, but it simplifies our exposition considerably. Let $x, x'$ be a generic pair of elements from $\mathcal{X}$ and let $\rho_* = \langle x, x' \rangle$ denote their inner product. Alternatively, we may refer to $\rho_*$ as (cosine) similarity or correlation. Again for simplicity, we assume that $0 \le \rho_* < 1$; the case of negative $\rho_*$ is a trivial extension because of symmetry.

We aim at reducing the dimensionality of the given data set by means of a random projection, which is realized by sampling a random matrix $A$ of dimension $k$ by $d$ whose entries are i.i.d. $N(0, 1)$ (i.e., zero-mean Gaussian with unit variance). Applying $A$ to $\mathcal{X}$ yields $\mathcal{Z} = \{z_i\}_{i=1}^n \subset \mathbb{R}^k$ with $z_i = Ax_i$, $i \in [n]$. Subsequently, the projected data points $\{z_i\}_{i=1}^n$ are subject to scalar quantization. A $b$-bit scalar quantizer is parameterized by 1) thresholds $\mathbf{t} = (t_1, \ldots, t_{K-1})$ with $0 = t_0 < t_1 < \ldots < t_{K-1} < t_K = +\infty$ inducing a partitioning of the positive real line into $K = 2^{b-1}$ intervals $\{[t_{r-1}, t_r), r \in [K]\}$ and 2) a codebook $\mathcal{M} = \{\mu_1, \ldots, \mu_K\}$ with code $\mu_r$ representing interval $[t_{r-1}, t_r), r \in [K]$. Given $\mathbf{t}$ and $\mathcal{M}$, the scalar quantizer (or quantization map) is defined by

$$Q : \mathbb{R} \to \mathcal{M}^\pm := -\mathcal{M} \cup \mathcal{M}, \quad z \mapsto Q(z) = \text{sign}(z) \sum_{r=1}^K \mu_r I(|z| \in [t_{r-1}, t_r)) \tag{1}$$

The projected, $b$-bit quantized data result as $\mathcal{Q} = \{q_i\}_{i=1}^n \subset (\mathcal{M}^\pm)^k$, $q_i = (Q(z_{ij}))_{j=1}^k$, $i \in [n]$.

**Problem statement.** Let $z, z'$ and $q, q'$ denote the pairs corresponding to $x, x'$ in $\mathcal{Z}$ respectively $\mathcal{Q}$. The goal is to estimate $\rho_* = \langle x, x' \rangle$ from $q, q'$ which automatically yields an estimate of $\|x - x'\|_2^2 = 2(1 - \rho_*)$. If $z, z'$ were given, it would be standard to use $\frac{1}{k} \langle z, z' \rangle$ as an unbiased estimator of $\rho_*$. This "linear" approach is commonly adopted when the data undergo uniform quantization with saturation level $T$ (i.e., $t_r = T \cdot r/(K-1)$, $\mu_r = (t_r - t_{r-1})/2$, $r \in [K-1]$, $\mu_K = T$), based on the rationale that as $b \to \infty$, $\frac{1}{k} \langle q, q' \rangle \to \frac{1}{k} \langle z, z' \rangle$ which in turn is sharply concentrated around its expectation $\rho_*$.

There are two major concerns about this approach. First, for finite $b$ the estimator $\frac{1}{k} \langle q, q' \rangle$ has a bias resulting from the non-linearity of $Q$ that does not vanish as $k \to \infty$. For small $b$, the effect of this bias is particularly pronounced. Lloyd-Max quantization (see Proposition 1 below) in place of

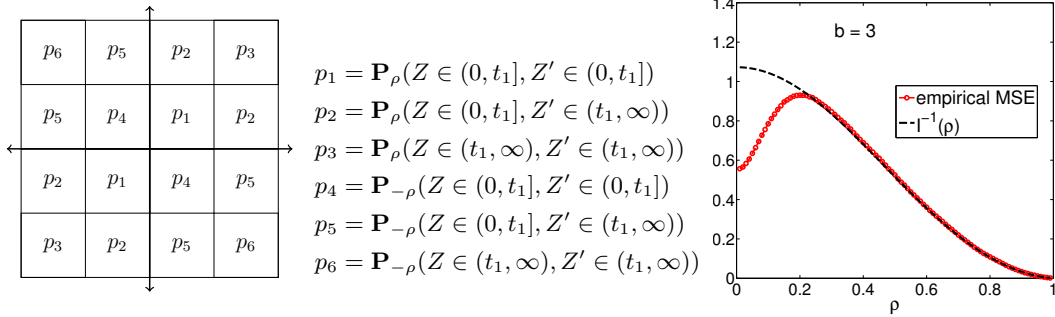

Figure 1: (L, M): Partitioning into cells for $b = 2$ and cell probabilities. (R): Empirical MSE $k(\widehat{\rho}_{\text{MLE}} - \rho_*)^2$ for $b = 3$ (averaged over $10^4$ i.i.d. data sets with $k = 100$) compared to the inverse information. The disagreement for $\rho \leq 0.2$ results from positive truncation of the MLE at zero.

uniform quantization provides some remedy, but the issue of non-vanishing bias remains. Second, even for infinite $b$, the approach is statistically not efficient. In order to see this, note that

$$\{(z_j, z_j')\}_{j=1}^{k} \overset{\text{i.i.d.}}{\sim} (Z, Z'), \quad \text{where } (Z, Z') \sim N_2\left(0, \begin{pmatrix} 1 & \rho_* \\ \rho_* & 1 \end{pmatrix}\right). \tag{2}$$

It is shown in [16] that the MLE of $\rho_*$ under the above bivariate normal model has a variance of $(1 - \rho_*^2)^2 / \{k(1 + \rho_*^2)\}$, while $\mathbf{Var}(\langle z, z' \rangle / k) = (1 + \rho_*^2)/k$ which is a substantial difference for large $\rho_*$. The higher variance results from not using the information that the components of $z$ and $z'$ have unit variance [16]. In conclusion, the linear approach as outlined above suffers from noticeable bias/and or high variance if the similarity $\rho_*$ is high, and it thus makes sense to study alternatives.

**Maximum likelihood estimation of $\rho_*$.** We here propose the MLE in place of the linear approach. The advantage of the MLE is that it can have substantially better statistical performance as the quantization map is explicitly taken into account. The MLE is based on bivariate normality according to (2). The effect of quantization is identical to that of what is known as interval censoring in statistics, i.e., in place of observing a specific value, one only observes that the datum is contained in an interval. The concept is easiest to understand in the case of one-bit quantization. For any $j \in [k]$, each of the four possible outcomes of $(q_j, q_j')$ corresponds to one of the four orthants of $\mathbb{R}^2$. By symmetry, the probability of $(q_j, q_j')$ falling into the positive or into the negative orthant are identical; both correspond to a "collision", i.e., to the event $\{q_j = q_j'\}$. Likewise, the probability of $(q_j, q_j')$ falling into one of the remaining two orthants are identical, corresponding to a disagreement $\{q_j \neq q_j'\}$. Accordingly, the likelihood function in $\rho$ is given by

$$\prod_{j=1}^{k} \{\pi(\rho)^{I(q_j = q_j')}(1 - \pi(\rho))^{I(q_j \neq q_j')}\}, \quad \pi(\rho) := \mathbf{P}_\rho(\text{sign}(Z) = \text{sign}(Z')),$$

where $\pi(\rho)$ denotes the probability of a collision after quantization for $(Z, Z')$ as in (2) with $\rho_*$ replaced by $\rho$. It is straightforward to show that the MLE is given by $\widehat{\rho}_{\text{MLE}} = \cos(\pi(1 - \widehat{\pi}))$, where $\pi$ is the circle constant and $\widehat{\pi} = k^{-1} \sum_{j=1}^{k} I(q_j = q_j')$ is the empirical counterpart to $\pi(\rho)$. We note that the expression for $\widehat{\rho}_{\text{MLE}}$ follows the same rationale as used for the simhash in [4].

With these preparations, it is not hard to see how the MLE generalizes to cases with more than one bit. For $b = 2$, there is a single non-trivial threshold $t_1$ that yields a partitioning of the real axis into four bins and accordingly a component $(q_j, q_j')$ of a quantized pair can fall into 16 possible cells (rectangles), cf. Figure 1. By orthant symmetry and symmetries within each orthant, one ends up with six distinct probabilities $p_1, \ldots, p_6$ for $(q_j, q_j')$ falling into one of those cells depending on $\rho$. Weighting those probabilities according to the number of their occurrences in the left part of Figure 1, we end up with probabilities $\pi_1 = \pi_1(\rho), \ldots, \pi_6 = \pi_6(\rho)$ that sum up to one. The corresponding relative cell frequencies $\widehat{\pi}_1, \ldots, \widehat{\pi}_6$ resulting from $(q_j, q_j')_{j=1}^{k}$ form a sufficient statistic for $\rho$. For general $b$, we have $2^{2b}$ cells and $L = K(K + 1)$ (recall that $K = 2^{b-1}$) distinct probabilities, so that $L = 20, 72, 272, 1056$ for $b = 3, \ldots, 6$. This yields the following compact expressions for the

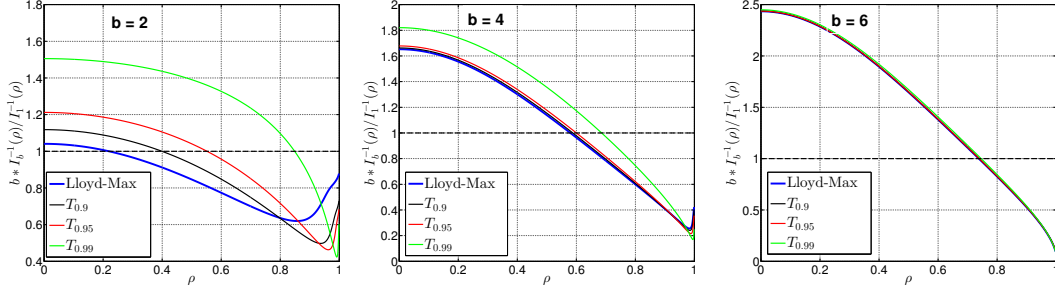

Figure 2: $b \cdot I_b^{-1}(\rho)/I_1^{-1}(\rho)$ vs. $\rho$ for different choices of $\mathbf{t}$: Lloyd-Max and uniform quantization with saturation levels $T_{0.9}, T_{0.95}, T_{0.99}$, cf. §4.1 for a definition. The latter are better suited for high similarity. The differences become smaller as $b$ increases. Note that for $b = 6$, $\rho > 0.7$ is required for either quantization scheme to achieve a better trade-off than the one-bit MLE.

negative log-likelihood $l(\rho)$ and the Fisher information $I(\rho) = \mathbf{E}_\rho[\ddot{l}(\rho)]$ (up to a factor of $k$)

$$l(\rho) = \sum_{\ell=1}^{L} \widehat{\pi}_\ell \log(\pi_\ell(\rho)), \quad I(\rho) = \sum_{\ell=1}^{L} \frac{(\dot{\pi}_\ell(\rho))^2}{\pi_\ell(\rho)}. \tag{3}$$

The information $I(\rho)$ is of particular interest. By classical statistical theory [21], $\{\mathbf{E}[\widehat{\rho}_{\text{MLE}}] - \rho_*\}^2 = O(1/k^2)$, $\mathbf{Var}(\widehat{\rho}_{\text{MLE}}) = I^{-1}(\rho)/k$, $\mathbf{E}[(\widehat{\rho}_{\text{MLE}} - \rho_*)^2] = I^{-1}(\rho)/k + O(1/k^2)$ as $k \to \infty$. While this is an asymptotic result, it agrees to a good extent with what one observes for finite, but not too small samples, cf. Figure 1. We therefore treat the inverse information as a proxy for the accuracy of $\widehat{\rho}_{\text{MLE}}$ in subsequent analysis.

*Remark.* We here briefly address the case of known, but possibly non-unit norms, i.e., $\|x\|_2 = \sigma_x$, $\|x'\|_2 = \sigma_{x'}$. This can be handled by re-scaling the thresholds of the quantizer (1) by $\sigma_x$ resp. $\sigma_{x'}$, estimating $\rho_*$ based on $q, q'$ as in the unit norm case, and subsequently re-scaling the estimate by $\sigma_x \sigma_{x'}$ to obtain an estimate of $\langle x, x' \rangle$. The assumption that the norms are known is not hard to satisfy in practice as they can be computed by one linear scan during data collection. With a limited bit budget, the norms additionally need to be quantized. It is unclear how to accurately estimate them from quantized data (for $b = 1$, it is definitely impossible).

**Choice of the quantizer.** Equipped with the Fisher information (3), one of the questions that can be addressed is quantizer design. Note that as opposed to the linear approach, the specific choice of the $\{\mu_r\}_{r=1}^{K}$ in (1) is not important as ML estimation only depends on cell frequencies but not on the values associated with the intervals $\{(t_{r-1}, t_r)\}_{r=1}^{K}$. The thresholds $\mathbf{t}$, however, turn out to have a considerable impact, at least for small $b$. An optimal set of thresholds can be determined by minimizing the inverse information $I^{-1}(\rho; \mathbf{t})$ w.r.t. $\mathbf{t}$ for fixed $\rho$. As the underlying similarity is not known, this may not seem practical. On the other hand, prior knowledge about the range of $\rho$ may be available, or the closed form one-bit estimator can be used as pilot estimator. For $\rho = 0$, the optimal set of thresholds coincide with those of Lloyd-Max quantization [20].

**Proposition 1.** *Let $g \sim N(0, 1)$ and consider Lloyd-Max quantization given by*

$$(\mathbf{t}^*, \{\mu_r^*\}_{r=1}^{K}) = \underset{\mathbf{t}, \{\mu_r\}_{r=1}^{K}}{\operatorname{argmin}} \mathbf{E}[\{g - Q(g; \mathbf{t}, \{\mu_r\}_{r=1}^{K})\}^2]. \text{ We also have } \mathbf{t}^* = \underset{\mathbf{t}}{\operatorname{argmin}} I^{-1}(0; \mathbf{t}).$$

The Lloyd-Max problem can be solved numerically by means of an alternating scheme which can be shown to converge to a global optimum [13]. For $\rho > 0$, an optimal set of thresholds can be determined by general procedures for nonlinear optimization. Evaluation of $I^{-1}(\rho; \mathbf{t})$ requires computation of the probabilities $\{\pi_\ell(\rho; \mathbf{t})\}_{\ell=1}^{L}$ and their derivatives $\{\dot{\pi}_\ell(\rho; \mathbf{t})\}_{\ell=1}^{L}$. The latter are available in closed form (cf. supplement), while for the former specialized numerical integration procedures [8] can be used. In order to avoid multi-dimensional optimization, it makes sense to confine oneself to thresholds of the form $t_r = T \cdot r/(K - 1)$, $r \in [K - 1]$, so that only $T$ needs to be optimized. Even though the Lloyd-Max scheme performs reasonably also for large values of $\rho$, the one-parameter scheme may still yield significant improvements in that case, cf. Figure 2. Once $b \geq 5$, the differences between the two schemes become marginal.

**Trade-off between $k$ and $b$.** Suppose we are given a fixed budget of bits $B = k \cdot b$ for transmission or storage, and we are in free choosing $b$. The optimal choice of $b$ can be determined by comparing

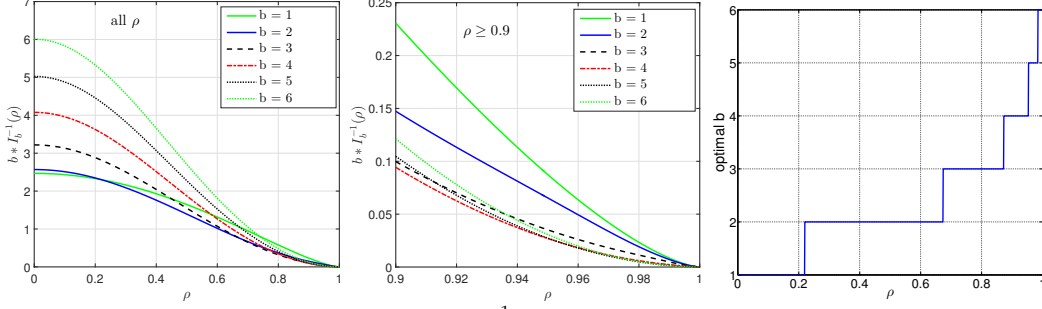

Figure 3: Trade-off between $k$ and $b$. (L): $b \cdot I_b^{-1}(\rho)$ vs. $\rho$ for $1 \leq b \leq 6$ with $\mathbf{t}$ chosen by Lloyd-Max. (M): Zoom into the range $0.9 \leq \rho \leq 1$. (R): choice of $b$ minimizing $b \cdot I_b^{-1}(\rho)$ vs. $\rho$.

the inverse Fisher information $I_b^{-1}(\rho)$ for changing $b$ with $\mathbf{t}$ chosen according to either of the two schemes above. Since the mean squared error of $\widehat{\rho}_{\mathrm{MLE}}$ decays with $1/k$ for any $b$, for $b'$ with $b' > b$ to be more efficient than $b$ at the bit scale it, is required that $I_{b'}(\rho)/I_b(\rho) > b'/b$ as with the smaller choice $b$ one would be allowed to increase $k$ by a factor of $b'/b$. Again, this comparison is dependent on a specific $\rho$. From Figure 3, however, one can draw general conclusions: for $\rho < 0.2$, it does not pay off to increase $b$ beyond one; as $\rho$ increases, higher values of $b$ achieve a better trade-off with even $b = 6$ being the optimal choice for $\rho > 0.98$. The intuition is that two points of high similarity agree on their first significant bit for most coordinates, in which case increasing the number of bits becomes beneficial. This finding is particularly relevant to (near-)duplicate detection/nearest neighbor search where high similarities prevail, an application investigated in §4.

**Rate of growth of the Fisher information near $\rho = 1$.** Interestingly, we do not observe a "saturation" even for $b = 6$ in the sense that for $\rho$ close enough to 1, one can still achieve an improvement at the bit scale compared to $1 \leq b \leq 5$. This raises the question about the rate of growth of the Fisher information near one relative to the full precision case ($b \to \infty$). As shown in [16] $I_\infty(\rho) = (1 + \rho^2)/(1 - \rho^2)^2 = \Theta((1 - \rho)^{-2})$ as $\rho \to 1$. As stated below, in the finite bit case, the exponent is only $3/2$ for all $b$. This is a noticeable gap.

**Theorem 1.** *For $1 \leq b < \infty$, we have $I(\rho) = \Theta((1 - \rho)^{-3/2})$ as $\rho \to 1$.*

The theorem has an interesting implication with regard to the existence of a Johnson-Lindenstrauss (J-L)-type result for quantized random projections. In a nutshell, the J-L lemma states that as long as $k = \Omega(\log n/\varepsilon^2)$, with high probability we have that

$$(1 - \varepsilon)\|x_i - x_j\|_2^2 \leq \|z_i - z_j\|_2^2/k \leq (1 + \varepsilon)\|x_i - x_j\|_2^2 \text{ for all pairs } (i, j),$$

i.e., the distances of the data in $\mathcal{X}$ are preserved in $\mathcal{Z}$ up to a relative error of $\varepsilon$. In our setting, one would hope for an equivalent of the form

$$(1 - \varepsilon)2(1 - \rho_{ij}) \leq 2(1 - \widehat{\rho}_{\mathrm{MLE}}^{ij}) \leq (1 + \varepsilon)2(1 - \rho_{ij}) \ \forall(i, j) \text{ as long as } k = \Omega(\log n/\varepsilon^2), \quad (4)$$

where $\rho_{ij} = \langle x_i, x_j \rangle$, $i, j \in [n]$, and $\widehat{\rho}_{\mathrm{MLE}}^{ij}$ denotes the MLE for $\rho_{ij}$ given quantized RPs. The standard proof of the J-L lemma [6] combines norm preservation for each individual pair of the form

$$\mathbf{P}((1 - \varepsilon)\|x_i - x_j\|_2^2 \leq \|z_i - z_j\|_2^2/k \leq (1 + \varepsilon)\|x_i - x_j\|_2^2) \leq 2\exp(-k\Theta(\varepsilon^2))$$

with a union bound. Such a concentration result does not appear to be attainable for $\widehat{\rho}_{\mathrm{MLE}} - \rho_*$, not even asymptotically as $k \to \infty$ in which case $\widehat{\rho}_{\mathrm{MLE}} - \rho_*$ is asymptotically normal with mean zero and variance $I^{-1}(\rho_*)/k$. This yields an asymptotic tail bound of the form

$$\mathbf{P}(|\widehat{\rho}_{\mathrm{MLE}} - \rho_*| > \delta) \leq 2\exp(-\delta^2 k/\{2I^{-1}(\rho_*)\}). \quad (5)$$

For a result of the form (4), which is about relative distance preservation, one would need to choose $\delta$ proportional to $\varepsilon(1 - \rho_*)$. In virtue of Theorem 1, $I^{-1}(\rho_*) = \Theta((1 - \rho_*)^{3/2})$ as $\rho_* \to 1$ so that with $\delta$ chosen in that way the exponent in (5) would vanish as $\rho_* \to 1$. By constrast, the required rate of decay of $I^{-1}(\rho_*)$ is achieved in the full precision case. Given the asymptotic optimality of the MLE according to the Cramer-Rao lower bound suggests that a qualitative counterpart to the J-L lemma (4) is out of reach. Weaker versions in which the required lower bound on $k$ would depend inversely on the minimum distance of points in $\mathcal{X}$ are still possible. Similarly, a weaker result of the form

$$2(1 - \rho_{ij}) - \varepsilon \leq 2(1 - \widehat{\rho}_{\mathrm{MLE}}^{ij}) \leq 2(1 - \rho_{ij}) + \varepsilon \ \forall(i, j) \text{ as long as } k = \Omega(\log n/\varepsilon^2),$$

is known to hold already in the one-bit case and follows immediately from the closed form expression of the MLE, Hoeffdings's inequality, and the union bound; cf. e.g. [10].

# 3 A general class of estimators and approximate MLE computation

A natural concern about the MLE relative to the linear approach is that it requires optimization via an iterative scheme. The optimization problem is smooth, one-dimensional and over the unit interval, hence not challenging for modern solvers. However, in applications it is typically required to compute the MLE many times, hence avoiding an iterative scheme for optimization is worthwhile. In this section, we introduce an approximation to the MLE that only requires at most two table look-ups.

**A general class of estimators.** Let $\pi(\rho) = (\pi_1(\rho), \ldots, \pi_L(\rho))^\top$, $\sum_{\ell=1}^{L} \pi_\ell(\rho) = 1$, be the normalized cell frequencies depending on $\rho$ as defined in §2, let further $w \in \mathbb{R}^L$ be a fixed vector of weights, and consider the map $\rho \mapsto \theta(\rho; w) := \langle \pi(\rho), w \rangle$. If $\langle \dot{\pi}(\rho), w \rangle > 0$ uniformly in $\rho$ (such $w$ always exist), $\theta(\cdot; w)$ is increasing and has an inverse $\theta^{-1}(\cdot; w)$. We can then consider the estimator

$$\widehat{\rho}_w = \theta^{-1}(\langle \widehat{\pi}, w \rangle; w), \tag{6}$$

where we recall that $\widehat{\pi} = (\widehat{\pi}, \ldots, \widehat{\pi}_L)^\top$ are the empirical cell frequencies given quantized data $q, q'$. It is easy to see that $\widehat{\rho}_w$ is a consistent estimator of $\rho_*$: we have $\widehat{\pi} \to \pi(\rho_*)$ in probability by the law of large numbers, and $\theta^{-1}(\langle \widehat{\pi}, w \rangle; w) \to \theta^{-1}(\langle \pi(\rho_*), w \rangle; w) = \theta^{-1}(\theta(\rho_*; w); w) = \rho_*$ by two-fold application of the continuous mapping theorem. By choosing $w$ such that $w_\ell = 1$ for $\ell$ corresponding to cells contained in the positive/negative orthant and $w_\ell = -1$ otherwise, $\widehat{\rho}_w$ becomes the one-bit MLE. By choosing $w_\ell = 1$ for diagonal cells (cf. Figure 1) corresponding to a collision event $\{q_j = q'_j\}$ and $w_\ell = 0$ otherwise, we obtain the Hamming distance-based estimator in [17]. Alternatively, we may choose $w$ such that the asymptotic variance of $\widehat{\rho}_w$ is minimized.

**Theorem 2.** *For any $w$ s.t. $\dot{\pi}(\rho_*)^\top w \neq 0$, we have $\mathbf{Var}(\widehat{\rho}_w) = V(w; \rho_*)/k + O(1/k^2)$ as $k \to \infty$,*

$$V(w; \rho_*) = (w^\top \Sigma(\rho_*) w)/\{\dot{\pi}(\rho_*)^\top w\}^2, \quad \Sigma(\rho_*) := \Pi(\rho_*) - \pi(\rho_*)\pi(\rho_*)^\top,$$

*and $\Pi(\rho_*) := diag(\, (\pi_\ell(\rho_*))_{\ell=1}^{L} \,)$. Moreover, let $w^* = \Pi^{-1}(\rho_*)\dot{\pi}(\rho_*)$. Then:*

$$\mathrm{argmin}_w V(w; \rho_*) = \{\alpha(w^* + c\mathbf{1}), \ \alpha \neq 0, c \in \mathbb{R}\}, \quad V(w^*; \rho_*) = I^{-1}(\rho_*),$$

*and $\mathbf{E}[(\widehat{\rho}_{w^*} - \rho_*)^2] = \mathbf{E}[(\widehat{\rho}_{MLE} - \rho_*)^2] + O(1/k^2)$.*

Theorem 2 yields an expression for the optimal weights $w^* = \Pi^{-1}(\rho_*)\dot{\pi}(\rho_*)$. This optimal choice is unique up to translation by a multiple of the constant vector $\mathbf{1}$ and scaling. The estimator $\widehat{\rho}_{w^*}$ based on the choice $w = w^*$ achieves asymptotically the same statistical performance as the MLE.

**Approximate computation.** The estimator $\widehat{\rho}_{w^*}$ is not operational as the optimal choice of the weights depends on the estimand itself. This issue can be dealt with by using a pilot estimator $\widehat{\rho}_0$ like the one-bit MLE, the Hamming distance-based estimator in [17] or $\widehat{\rho}_0 = \widehat{\rho}_{\overline{w}}$, where $\overline{w} = \int_0^1 w(\rho) \, d\rho$ averages the expression $w(\rho) = \Pi^{-1}(\rho)\dot{\pi}(\rho)$ for the optimal weights over $\rho$. Given the pilot estimator, we may then replace $w^*$ by $w(\widehat{\rho}_0)$ and use $\widehat{\rho}_{w(\widehat{\rho}_0)}$ as a proxy for $\widehat{\rho}_{w^*}$ which achieves the same statistical performance asymptotically.

A second issue is that computation of $\widehat{\rho}_w$ (6) entails inversion of the function $\theta(\cdot; w)$. The inverse may not be defined in general, but for the choices of $w$ that we have in mind, this is not a concern (cf. supplement). Inversion of $\theta(\cdot; w)$ can be carried out with tolerance $\varepsilon$ by tabulating the function values on a uniform grid of cardinality $\lceil 1/\varepsilon \rceil$ and performing a table lookup for each query. When computing $\widehat{\rho}_{w(\widehat{\rho}_0)}$, the weights depends on the data via the pilot estimator. We thus need to tabulate $w(\rho)$ on a grid, too. Accordingly, a whole set of look-up tables is required for function inversion, one for each set of weights. Given parameters $\varepsilon, \delta > 0$, a formal description of our scheme is as follows.

1. Set $R = \lceil 1/\varepsilon \rceil$, $\rho_r = r/R$, $r \in [R]$, and $B = \lceil 1/\delta \rceil$, $\rho_b = b/B$, $b \in [B]$.
2. Tabulate $w(\rho_b)$, $b \in [B]$, and function values $\theta(\rho_r; w(\rho_b)) = \langle w(\rho_b), \pi(\rho_r) \rangle$, $r \in [R]$, $b \in [B]$.

Steps 1. and 2. constitute a one-time pre-processing. Given data $q, q'$, we proceed as follows.

3. Obtain $\widehat{\pi}$ and the pilot estimator $\widehat{\rho}_0 = \theta^{-1}(\langle \widehat{\pi}, \overline{w} \rangle; \overline{w})$, with $\overline{w}$ defined in the previous paragraph.
4. Return $\widehat{\rho} = \theta^{-1}(\langle \widehat{\pi}, w(\widetilde{\rho}_0) \rangle; w(\widetilde{\rho}_0))$, where $\widetilde{\rho}_0$ is the value closest to $\widehat{\rho}_0$ among the $\{\rho_b\}$.

Step 2. requires about $C = \lceil 1/\varepsilon \rceil \cdot \lceil 1/\delta \rceil \cdot L$ computations/storage. From experimental results we find that $\varepsilon = 10^{-4}$ and $\delta = .02$ appear sufficient for practical purposes, which is still manageable even for $b = 6$ with $L = 1056$ cells in which case $C \approx 5 \times 10^8$. Again, this cost is occurred only once independent of the data. The function inversions in steps 3. and 4. are replaced by table lookups. By organizing computations efficiently, the frequencies $\widehat{\pi}$ can be obtained from one pass over $(q_j \cdot q'_j)$, $j \in [k]$. Equipped with the look-up tables, estimating the similarity of two points requires $O(k + L + \log(1/\varepsilon))$ flops which is only slightly more than a linear scheme with $O(k)$.

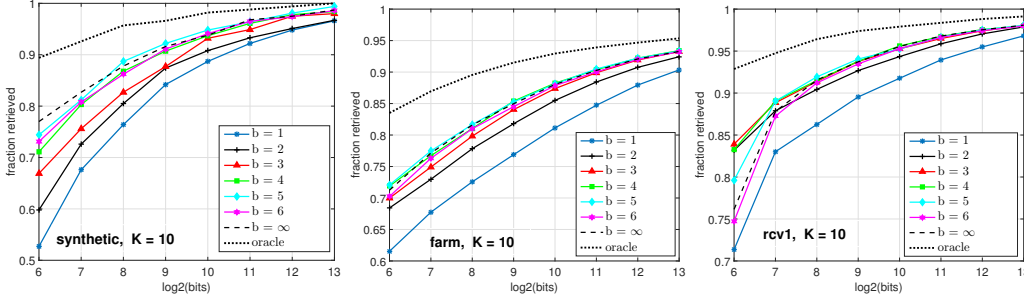

Figure 4: Average fraction of $K = 10$ nearest neighbors retrieved vs. total # of bits ($\log_2$ scale) for $1 \leq b \leq 6$. $b = \infty$ (dashed) represents the MLE based on unquantized data, with $k$ as for $b = 6$. The oracle curve (dotted) corresponds to $b = \infty$ with maximum $k$ (i.e., as for $b = 1$).

## 4   Experiments

We here illustrate the approach outlined above in nearest neighbor search and linear classification. The focus is on the trade-off between $b$ and $k$, in particular in the presence of high similarity.

### 4.1 Nearest Neighbor Search

Finding the most similar data points for a given query is a standard task in information retrieval. Another application is nearest neighbor classification. We here investigate how the performance of our approach is affected by the choice of $k$, $b$ and the quantization scheme. Moreover, we compare to two baseline competitors, the Hamming distance-based approach in [17] and the linear approach in which the quantized data are treated like the original unquantized data. For the approach in [17], similarity of the quantized data is measured in terms of their Hamming distance $\sum_{j=1}^{k} I(q_j \neq q'_j)$.

*Synthetic data.*   We generate $k$ i.i.d. samples of Gaussian data, where each sample $X = (X_0, X_1, \ldots, X_{96})$ is generated as $X_0 \sim N(0, 1)$, $X_j = \rho_j X_0 + (1 - \rho_j^2)^{1/2} Z_j$, $1 \leq j \leq 96$, where the $\{Z_j\}_{j=1}^{96}$ are i.i.d. $N(0, 1)$ and independent of $X_0$. We have $\mathbf{E}[(X_0 - X_j)^2] = 2(1 - \rho_j)$, where $\rho_j = \min\{0.8 + (j-1)0.002, 0.99\}$, $1 \leq j \leq 96$. The thus generated data subsequently undergo $b$-bit quantization, for $1 \leq b \leq 6$. Regarding the number of samples, we let $k \in \{2^6/b, 2^7/b, \ldots, 2^{13}/b\}$ which yields bit budgets between $2^6$ and $2^{13}$ for all $b$. The goal is to recover the $K$ nearest neighbors of $X_0$ according to the $\{\rho_j\}$, i.e., $X_{96}$ is the nearest neighbor etc. The purpose of this specific setting is to mimic the use of quantized random projections in the situation of a query $x_0$ and data points $\mathcal{X} = \{x_1, \ldots, x_{96}\}$ having cosine similarities $\{\rho_j\}_{j=1}^{96}$ with the query.

*Real data.*   We consider the Farm Ads data set ($n = 4{,}143$, $d = 54{,}877$) from the UCI repository and the RCV1 data set ($n = 20{,}242$, $d = 47{,}236$) from the LIBSVM webpage [3]. For both data sets, each instance is normalized to unit norm. As queries we select all data points whose first neighbor has (cosine) similarity less than $0.999$, whose tenth neighbor has similarity at least $0.8$ and whose hundredth neighbor has similarity less than $0.5$. These restrictions allow for a more clear presentation of our results. Prior to nearest neighbor search, $b$-bit quantized random projections are applied to the data, where the ranges for $b$ and for the number of projections $k$ is as for the synthetic data.

*Quantization.*   Four different quantization schemes are considered: Lloyd-Max quantization and thresholds $t_r = T_\rho \cdot r/(K-1)$, $r \in [K-1]$, where $T_\rho$ is chosen to minimize $I^{-1}(\rho)$; we consider $\rho \in \{0.9, 0.95, 0.99\}$. For the linear approach, we choose $\mu_r = \mathbf{E}[g | g \in (t_{r-1}, t_r)]$, $r \in [K]$, where $g \sim N(0, 1)$. For our approach and that in [17] the specific choice of the $\{\mu_r\}$ is not important.

*Evaluation.*   We perform 100 respectively 20 independent replications for synthetic respectively real data. We then inspect the top $K$ neighbors for $K \in \{3, 5, 10\}$ returned by the methods under consideration, and for each $K$ we report the average fraction of true $K$ neighbors that have been retrieved over 100 respectively 20 replications, where for the real data, we also average over the chosen queries (366 for farm and 160 for RCV1).

The results of our experiments point to several conclusions that can be summarized as follows. One-bit quantization is consistently outperformed by higher-bit quantization. The optimal choice of $b$ depends on the underlying similarities, and interacts with the choice of $\mathbf{t}$. It is an encouraging result that the performance based on full precision data (with $k$ as for $b = 6$) can essentially be matched

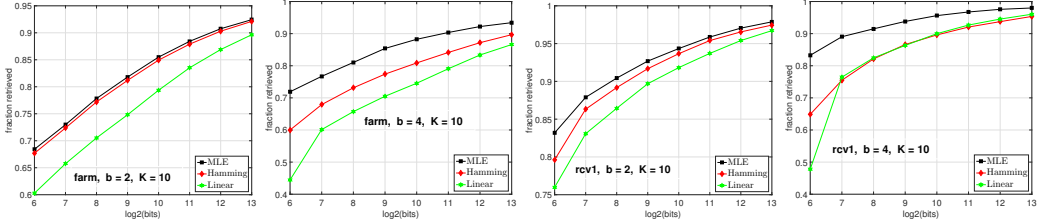

Figure 5: Average fraction of $K = 10$ nearest neighbors retrieved vs. total # of bits ($\log_2$ scale) of our approach (MLE) relative to that based on the Hamming distance and the linear approach for $b = 2, 4$.

when quantized data is used. For $b = 2$, the performance of the MLE is only marginally better than the approach based on the Hamming distance. The superiority of the former becomes apparent once $b \geq 4$ which is expected since for increasing $b$ the Hamming distance is statistically inefficient as it only uses the information whether a pair of quantized data agrees/disagrees. Some of these findings are reflected in Figures 4 and 5. We refer to the supplement for additional figures.

### 4.2 Linear Classification

We here outline an application to linear classification given features generated by (quantized) random projections. We aim at reconstructing the original Gram matrix $G = (\langle x_i, x'_i \rangle)_{1 \leq i, i' \leq n}$ from $\widehat{G} = (\widehat{g}_{ii'})$, where for $i \neq i'$, $\widehat{g}_{ii'} = \widehat{\rho}_{\text{MLE}}(q_i, q_{i'})$ equals the MLE of $\langle x_i, x'_i \rangle$ given a quantized data pair $q_i, q'_i$, and $\widehat{g}_{ii'} = 1$ else (assuming normalized data). The matrix $\widehat{G}$ is subsequently fed into LIBSVM. For testing, the inner products between test and training pairs are approximated accordingly.

*Setup.* We work with the farm data set using the first 3,000 samples for training, and the Arcene data set from the UCI repository with 100 training and 100 test samples in dimension $d = 10^4$. The choice of $k$ and $b$ is as in §4.1; for arcene, the total bit budget is lowered by a factor of 2. We perform 20 independent replications for each combination of $k$ and $b$. For SVM classification, we consider logarithmically spaced grids between $10^{-3}$ and $10^3$ for the parameter $C$ (cf. LIBSVM manual).

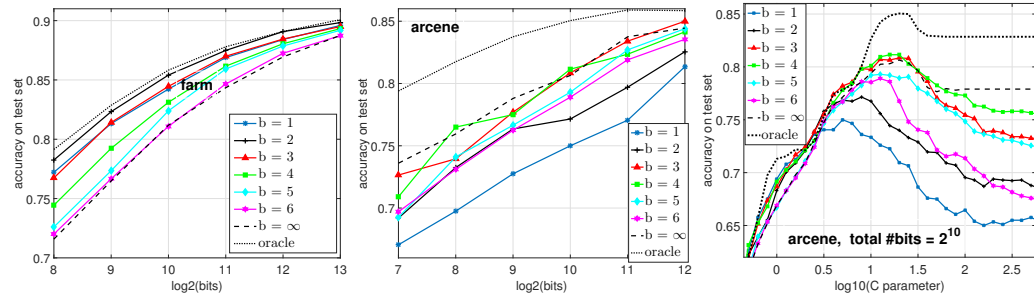

Figure 6: (L, M): accuracy vs. bits, optimized over the SVM parameter $C$. (R) accuracy vs. $C$ for a fixed # bits. $b = \infty$ indicates the performance based on unquantized data with $k$ as for $b = 6$. The oracle curve (dotted) corresponds to $b = \infty$ with maximum $k$ (i.e., as for $b = 1$).

Figure 6 (L, M) displays the average accuracy on the test data (after optimizing over $C$) in dependence of the bit budget. For the farm Ads data set, $b = 2$ achieves the best trade-off, followed by $b = 1$ and $b = 3$. For the Arcene data set, $b = 3, 4$ is optimal. In both cases, it does not pay off to go for $b \geq 5$.

## 5 Conclusion

In this paper, we bridge the gap between random projections with full precision and random projections quantized to a single bit. While Theorem 1 indicates that an exact counterpart to the J-L lemma is not attainable, other theoretical and empirical results herein point to the usefulness of the intermediate cases which give rise to an interesting trade-off that deserves further study in contexts where random projections can naturally be applied e.g. linear learning, nearest neighbor classification or clustering. The optimal choice of $b$ eventually depends on the application: increasing $b$ puts an emphasis on local rather than global similarity preservation.

**Acknowledgement**

The work of Ping Li and Martin Slawski is supported by NSF-Bigdata-1419210 and NSF-III-1360971. The work of Michael Mitzenmacher is supported by NSF CCF-1535795 and NSF CCF-1320231.

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
