[Supplementary Material]

# Supplement to 'Quantized Random Projections and Non-Linear Estimation of Cosine Similarity'

**Ping Li**
Rutgers University
pingli@stat.rutgers.edu

**Michael Mitzenmacher**
Harvard University
michaelm@eecs.harvard.edu

**Martin Slawski**
Rutgers University
martin.slawski@rutgers.edu

## Part I: Proofs

## A  Preparations

Let us recall the definition of the quantization map in the paper.

A $b$-bit scalar quantizer is parameterized by 1) thresholds $\mathbf{t} = (t_1, \ldots, t_{K-1})$ with $0 = t_0 < t_1 < \ldots < t_{K-1} < t_K = +\infty$ inducing a partitioning of the positive real line into $K = 2^{b-1}$ intervals $\{[t_{r-1}, t_r), \ r \in [K]\}$ and 2) a codebook $\mathcal{M} = \{\mu_1, \ldots, \mu_K\}$ with code $\mu_r$ representing interval $[t_{r-1}, t_r), r \in [K]$. Given $\mathbf{t}$ and $\mathcal{M}$, the scalar quantizer (or quantization map) is defined by

$$Q : \mathbb{R} \to \mathcal{M}^{\pm} := -\mathcal{M} \cup \mathcal{M}, \quad z \mapsto Q(z) = \text{sign}(z) \sum_{r=1}^{K} \mu_r I(|z| \in [t_{r-1}, t_r)). \qquad (1)$$

It is often convenient to work with an expanded set of thresholds

$$\tau_1 = -t_K, \tau_2 = -t_{K-1}, \ldots, \tau_K = -t_1, \tau_{K+1} = 0, \tau_{K+2} = t_1, \ldots, \tau_J = t_K, \quad J = 2K + 1.$$

and codes

$$\nu_1 = -\mu_K, \nu_2 = -\mu_{K-1}, \ldots, \nu_K = -\mu_1, \nu_{K+1} = \mu_1, \ldots, \nu_{2K} = \mu_K.$$

Consider a pair $q = (q_j)_{j=1}^k, q' = (q'_j)_{j=1}^k$ of quantized data after applying $Q$ component-wise to $z = (z_j)_{j=1}^k$ respectively $z' = (z'_j)_{j=1}^k$. Each pair $q_j, q'_j$ can be associated with a rectangle from the set $\{(\tau_r, \tau_{r+1}) \times (\tau_s, \tau_{s+1}), \ r, s \in [2K]\}$. This yields a partitioning of $\mathbb{R}^2$ into $2^{2b}$ rectangles (cf. Figure 1). In view of

$$\{(z_j, z'_j)\}_{j=1}^k \overset{\text{i.i.d.}}{\sim} (Z, Z'), \quad \text{where } (Z, Z') \sim N_2\left(0, \begin{pmatrix} 1 & \rho_* \\ \rho_* & 1 \end{pmatrix}\right) \qquad (2)$$

we have for $j \in [k], r, s \in [2K]$

$$\begin{aligned} \mathbf{P}(q_j = \nu_r, \ q'_j = \nu_s) &= \mathbf{P}_{\rho_*}\{z_j \in (\tau_r, \tau_{r+1}), z'_j \in (\tau_s, \tau_{s+1})\} \\ &= \int_{\tau_r}^{\tau_{r+1}} \int_{\tau_s}^{\tau_{s+1}} \frac{1}{2\pi\sqrt{1-\rho_*^2}} \exp\left(-\frac{(x^2 - 2\rho_* xy + y^2)}{2(1-\rho_*)^2}\right) \ dxdy, \end{aligned}$$

where here and below, $\mathbf{P}_\rho(A)$ denotes the probability of some event $A$ under the bivariate normal model (2) with $\rho_*$ replaced by an arbitrary $\rho \in (-1, 1)$.

While there are $2^{2b}$ possible rectangles, the associated probabilities as given in the above display have (at most) $L = K(K + 1)$ distinct values because of symmetries.

- We have $\mathbf{P}(q_j = \nu_r, \ q'_j = \nu_s) = \mathbf{P}(q_j = \nu_s, \ q'_j = \nu_r)$
- We have $\mathbf{P}(q_j = \nu_r, \ q'_j = \nu_s) = \mathbf{P}(q_j = -\nu_r, \ q'_j = -\nu_s)$

| $p_{2G}$ | $\cdots$ | $p_{G+2K-1}$ | $p_{G+K}$ | $p_K$ | $p_{2K-1}$ | $\cdots$ | $p_G$ |
|---|---|---|---|---|---|---|---|
| $\vdots$ | $\cdot^{\cdot^{\cdot}}$ | $\vdots$ | $\vdots$ | $\vdots$ | $\vdots$ | $\cdot^{\cdot^{\cdot}}$ | $\vdots$ |
| $p_{G+2K-1}$ | $\cdots$ | $p_{G+K+1}$ | $\vdots$ | $\vdots$ | $p_{K+1}$ | $\cdots$ | $p_{2K-1}$ |
| $p_{G+K}$ | $\cdots$ | $\cdots$ | $p_{G+1}$ | $p_1$ | $\cdots$ | $\cdots$ | $p_K$ |
| $p_K$ | $\cdots$ | $\cdots$ | $p_1$ | $p_{G+1}$ | $\cdots$ | $\cdots$ | $p_{G+K}$ |
| $p_{2K-1}$ | $\cdots$ | $p_{K+1}$ | $\vdots$ | $\vdots$ | $p_{G+K+1}$ | $\cdots$ | $p_{G+2K-1}$ |
| $\vdots$ | $\cdot^{\cdot^{\cdot}}$ | $\vdots$ | $\vdots$ | $\vdots$ | $\vdots$ | $\cdot^{\cdot^{\cdot}}$ | $\vdots$ |
| $p_G$ | $\cdots$ | $p_{2K-1}$ | $p_K$ | $p_{G+K}$ | $p_{G+2K-1}$ | $\cdots$ | $p_{2G}$ |

Figure 1: Distinct rectangle probabilities $p_1, \ldots, p_{2G}$, where $G = L/2$ is the number of distinct probabilities per quadrant.

for all $j \in [k]$, $r, s \in [2K]$. This is illustrated in Figure 1 (the corresponding figure in the paper shows the special case $b = 2$).

We also note that

$$\mathbf{P}_\rho(q_j = -\nu_r, \, q'_j = \nu_s) = \mathbf{P}_\rho(q_j = \nu_r, \, q'_j = -\nu_s) = \mathbf{P}_{-\rho}(q_j = \nu_r, \, q'_j = \nu_s) \qquad (3)$$

i.e. evaluating the probability under a sign flip for one of $(q_j, q'_j)$ can be done by flipping the sign of $\rho$ instead.

The cell probabilities as depicted in Figure 1 are given by

$$p_1(\rho) = \mathbf{P}_\rho\{Z \in (t_0, t_1), \, Z' \in (t_0, t_1)\},$$
$$p_2(\rho) = \mathbf{P}_\rho\{Z \in (t_0, t_1), \, Z' \in (t_1, t_2)\},$$
$$\vdots$$
$$p_K(\rho) = \mathbf{P}_\rho\{Z \in (t_0, t_1), \, Z' \in (t_{K-1}, t_K)\}$$
$$p_{K+1}(\rho) = \mathbf{P}_\rho\{Z \in (t_1, t_2), \, Z' \in (t_1, t_2)\},$$
$$\vdots$$
$$p_{2K-1}(\rho) = \mathbf{P}_\rho\{Z \in (t_1, t_2), \, Z' \in (t_{K-1}, t_K)\},$$
$$\vdots$$
$$p_G(\rho) = \mathbf{P}_\rho\{Z \in (t_{K-1}, t_K), Z' \in (t_{K-1}, t_K)\}.$$

In virtue of property (3), the remaining probabilities $p_{G+1}, \ldots, p_{2G} = p_L$ are given by

$$p_{G+g}(\rho) = p_g(-\rho), \ g = 1, \ldots, G. \tag{4}$$

The cells containing the angle bisector are referred to as 'diagonal cells' and the associated probabilities

$$p_1(\rho) = \mathbf{P}(q_j = \mu_1, q_j' = \mu_1), \quad p_{K+1}(\rho) = \mathbf{P}(q_j = \mu_2, q_j' = \mu_2), \ldots, \tag{5}$$

$$p_G(\rho) = \mathbf{P}(q_j = \mu_K, q_j' = \mu_K). \tag{6}$$

as 'collision probabilities' corresponding to the event $\{q_j = q_j'\}$.

We define weights $\omega_\ell$, $\ell \in [L]$ (recall that $L = 2G$), according to the number occurrences of the probabilities $p_\ell$, $\ell \in [L]$, in the table shown in Figure 1. We have $\omega_\ell = 2$ for the collision probabilities (5) and their counterparts (in the sense of (4)) $p_{G+1}(\rho), p_{G+K+1}(\rho), \ldots, p_{2G}(\rho)$, and $\omega_\ell = 4$ otherwise.

We are then in position to define

$$\pi_\ell = \pi_\ell(\rho) = p_\ell(\rho) \cdot \omega_\ell, \ \ell \in [L].$$

and $\pi = \pi(\rho) = (\pi_\ell(\rho))_{\ell=1}^L$, which defines a probability distribution over a reduced set of cells, merging those that have the same probability because of symmetry. In particular, $\sum_{\ell=1}^L \pi_\ell(\rho) = 1$ for all $\rho$. Dependence on $\rho$ is occasionally omitted.

For the proofs, we will make use of closed form expressions for the derivatives $\dot{\pi}_\ell(\rho)$, $\ell \in [L]$.

**Lemma A.1.** *Let $(Z, Z')$ follow a bivariate normal distribution as in* (2). *For any $a < b$, $u < v$, we have*

$$\frac{d \mathbf{P}_\rho(Z \in [a, b], Z' \in [u, v])}{d\rho}$$

$$= \frac{1}{2\pi\sqrt{1-\rho^2}} \left\{ \exp(-v^2/2) \left\{ \exp\left(-\frac{(b-v\rho)^2}{2(1-\rho^2)}\right) - \exp\left(-\frac{(a-v\rho)^2}{2(1-\rho^2)}\right) \right\} - \right.$$

$$\left. - \exp(-u^2/2) \left\{ \exp\left(-\frac{(b-u\rho)^2}{2(1-\rho^2)}\right) - \exp\left(-\frac{(a-u\rho)^2}{2(1-\rho^2)}\right) \right\} \right\}$$

*Proof.* The proof is along the lines of the proof of Lemma 1 in [1]. $\qquad\square$

# B   Proof of Proposition 1

We first re-arrange the optimization problem for finding the optimal set of thresholds yielding minimum inverse Fisher information:

$$\min_{\mathbf{t}} I^{-1}(0; \mathbf{t}).$$

We have

$$I^{-1}(0; \mathbf{t}) = \left( \sum_{\ell=1}^L \frac{\{\dot{\pi}_\ell(0; \mathbf{t})\}^2}{\pi_\ell(0; \mathbf{t})} \right)^{-1}$$

$$= \left( \sum_{r=1}^{2J} \sum_{s=1}^{2J} \frac{\left( \frac{d}{d\rho} \mathbf{P}_\rho\{Z \in (\tau_r, \tau_{r+1}), Z' \in (\tau_s, \tau_{s+1})\}\big|_{\rho=0} \right)^2}{\mathbf{P}_\rho\{Z \in (\tau_r, \tau_{r+1}), Z' \in (\tau_s, \tau_{s+1})\}} \right)^{-1}$$

$$= \frac{1}{4} \left( \sum_{r=1}^K \sum_{s=1}^K \frac{\left( \frac{d}{d\rho} \mathbf{P}_\rho\{Z \in [t_{r-1}, t_r), Z' \in [t_{s-1}, t_s)\}\big|_{\rho=0} \right)^2}{\mathbf{P}_\rho\{Z \in [t_{r-1}, t_r), Z' \in [t_{s-1}, t_s)\}} \right)^{-1}$$

In order to obtain the last identity, note that for $\rho = 0$ all quadrants as depicted in Figure 1 are exchangeable. It thus suffices to consider the cells in one quadrant.

By independence, for all pairs $(r, s)$

$$\mathbf{P}_\rho\{Z \in [t_{r-1}, t_r), Z' \in [t_{s-1}, t_s)\} = \mathbf{P}\{Z \in [t_{r-1}, t_r)\} \cdot \mathbf{P}\{Z' \in [t_{s-1}, t_s)\}$$
$$= (\Phi(t_r) - \Phi(t_{r-1}))(\Phi(t_s) - \Phi(t_{s-1})),$$

where $\Phi$ denotes the cdf of the $N(0,1)$-distribution. Moreover, invoking Lemma A.1 with $\rho$ set to zero and collecting terms

$$\frac{d}{d\rho} \mathbf{P}_\rho\{Z \in [t_{r-1}, t_r), Z' \in [t_{s-1}, t_s)\}\Big|_{\rho=0} = (\phi(t_r) - \phi(t_{r-1}))(\phi(t_s) - \phi(t_{s-1})),$$

where $\phi$ denotes the pdf of the $N(0,1)$-distribution. Accordingly, we consider

$$I^{-1}(0; \mathbf{t}) = \frac{1}{4} \left\{ \sum_{r=1}^{K} \sum_{s=1}^{K} \frac{(\phi(t_r) - \phi(t_{r-1}))^2}{(\Phi(t_r) - \Phi(t_{r-1}))} \frac{(\phi(t_s) - \phi(t_{s-1}))^2}{(\Phi(t_s) - \Phi(t_{s-1}))} \right\}^{-1}$$

Noting that for $g \sim N(0,1)$, for any $\alpha < \beta$

$$\phi(\beta) - \phi(\alpha) = \int_\alpha^\beta \phi'(z)\, dz = \int_\alpha^\beta -z\phi(z)\, dz = -\mathbf{E}[g | g \in (\alpha, \beta)]\, \mathbf{P}(g \in (\alpha, \beta)),$$

we obtain that

$$I^{-1}(0; \mathbf{t}) = \frac{1}{4} \left( \sum_{r=1}^{K} \mathbf{E}[g | g \in (t_{r-1}, t_r)]^2\, \mathbf{P}(g \in (t_{r-1}, t_r)) \right)^{-2}. \tag{7}$$

We now consider the Lloyd-Max problem, re-arrange it so as to show equivalence of the two optimization problems.

Lloyd-Max quantization is based on the optimization problem

$$(\mathbf{t}^*, \{\mu_r^*\}_{r=1}^K) = \operatorname*{argmin}_{\mathbf{t}, \{\mu_r\}_{r=1}^K} \mathbf{E}[\{g - Q(g; \mathbf{t}, \{\mu_r\}_{r=1}^K)\}^2], \quad \text{where } g \sim N(0,1),$$

and $Q$ as defined in (1). For the above problem, it is not hard to see that for any fixed choice of $\mathbf{t}$, the minimizing $\boldsymbol{\mu}^*(\mathbf{t})$ is given by $\mu_r^*(\mathbf{t}) = \mathbf{E}[g | g \in (t_{r-1}, t_r)]$, $r \in [K]$. To finish the proof, it thus remains to show that after substituting $\boldsymbol{\mu}^*(\mathbf{t})$ back, the resulting minimization problem in $\mathbf{t}$ is equivalent to minimizing $I^{-1}(0; \mathbf{t})$. We have

$$\min_{\mathbf{t}} \mathbf{E} \left[ \left\{ g - \operatorname{sign}(g) \sum_{r=1}^{K} I(|g| \in (t_{r-1}, t_r)) \mathbf{E}[g | g \in (t_{r-1}, t_r)] \right\}^2 \right]$$

$$= 2 \min_{\mathbf{t}} \mathbf{E} \left[ \sum_{r=1}^{K} I(g \in (t_{r-1}, t_r))(g - \mathbf{E}[g | g \in (t_{r-1}, t_r)])^2 \right]$$

$$= 2 \min_{\mathbf{t}} \mathbf{E} \left[ \sum_{r=1}^{K} I(g \in (t_{r-1}, t_r)) \left\{ g^2 - 2g\, \mathbf{E}[g | g \in (t_{r-1}, t_r)] + \mathbf{E}[g | g \in (t_{r-1}, t_r)]^2 \right\} \right]$$

$$= \mathbf{E}[g^2] + 2 \min_{\mathbf{t}} \left\{ -2 \sum_{k=1}^{K} \mathbf{E}[g | g \in (t_{r-1}, t_r)]\, \mathbf{E}[I(g \in (t_{r-1}, t_r))g] + \right.$$

$$\left. + \sum_{r=1}^{K} \mathbf{P}(g \in (t_{r-1}, t_r))\, \mathbf{E}[g | g \in (t_{r-1}, t_r)]^2 \right\}$$

$$= 1 + \min_{\mathbf{t}} -2 \sum_{r=1}^{K} \mathbf{E}[g | g \in (t_{r-1}, t_r)]^2\, \mathbf{P}(g \in (t_{r-1}, t_r))$$

Comparison with (7) shows that both optimization problems amount to maximization of

$$\sum_{r=1}^{K} \mathbf{E}[g | g \in (t_{r-1}, t_r)]^2\, \mathbf{P}(g \in (t_{r-1}, t_r)).$$

w.r.t. $\mathbf{t}$. This concludes the proof.

## C    Proof of Theorem 1

We start by extracting the factor $\frac{1}{2\pi}\frac{1}{\sqrt{1-\rho^2}}$ from the $\{\dot{\pi}_\ell(\rho)\}_{\ell\in[L]}$, so that we end up with terms $\{\gamma_\ell(\rho)\}_{\ell\in[L]}$ only involving exponentials (cf. Lemma A.1).

$$I(\rho) = \sum_{\ell=1}^{L} \frac{(\dot{\pi}_\ell(\rho))^2}{\pi_\ell(\rho)} = \frac{1}{4\pi^2}\frac{1}{1-\rho^2}\sum_{\ell=1}^{L}\frac{\gamma_\ell(\rho)^2}{\pi_\ell(\rho)}, \quad \gamma_\ell(\rho) = 2\pi\sqrt{1-\rho^2}\,\dot{\pi}_\ell(\rho),\ \ell\in[L]. \qquad (8)$$

We thus need to prove that the sum on the r.h.s. is $\Theta(1/\sqrt{1-\rho})$ as $\rho\to 1$ as this would yield $I(\rho) = \Theta((1-\rho)^{-3/2})$ as $\rho\to 1$. For this purpose, we consider that sum term by term.

- Note that $\pi_\ell(\rho) = \Theta(1)$ as $\rho\to 1$ only for those $\ell$ corresponding to diagonal cells (collision probabilities, cf. (5)). These terms eventually take the role of lower order terms.

- Accordingly, the remaining $\pi_\ell(\rho)$ vanish as $\rho\to 1$.

The latter terms thus deserve further investigation. By a first-order Taylor expansion of $\pi_\ell(\rho)$ around $\rho_0$:

$$\pi_\ell(\rho) - \pi_\ell(\rho_0) = (\rho-\rho_0)\dot{\pi}_\ell(\rho_0) + O((\rho-\rho_0)^2) \text{ as } |\rho-\rho_0|\to 0$$
$$= \frac{1}{2\pi}\frac{\rho-\rho_0}{\sqrt{1-\rho_0^2}}\gamma_\ell(\rho_0) + O((\rho-\rho_0)^2)$$

Consequently, for all $\ell$ such that $\pi_\ell(\rho) = o(1)$ as $\rho\to 1$, we have

$$\frac{\gamma_\ell(\rho)^2}{\pi_\ell(\rho)} = \Theta\left(\frac{\gamma_\ell(\rho)}{\sqrt{1-\rho}}\right) \text{ as } \rho\to 1. \qquad (9)$$

It turns out that apart from the diagonal cells (which can be ignored as explained above), only for those $\ell$ corresponding to cells directly adjacent to diagonal cells, it holds that $\gamma_\ell = \Theta(1)$; these constitute the leading terms which then yield the order $I(\rho) = \Theta((1-\rho)^{-3/2})$ as $\rho\to 1$. To make this argument explicit, let us consider the form of the $\gamma_\ell(\rho)$ in detail (cf. Lemma A.1).

$$\gamma_\ell(\rho) = 2\pi\sqrt{1-\rho^2}\omega_\ell\frac{d\mathbf{P}_\rho\{Z\in(\tau_r,\tau_{r+1}), Z'\in(\tau_s,\tau_{s+1})\}}{d\rho}$$
$$= \omega_\ell\left\{\exp(-\tau_{s+1}^2/2)\left\{\exp\left(-\frac{(\tau_{r+1}-\tau_{s+1}\rho)^2}{2(1-\rho^2)}\right) - \exp\left(-\frac{(\tau_r-\tau_{s+1}\rho)^2}{2(1-\rho^2)}\right)\right\} -\right.$$
$$\left. - \exp(-\tau_s^2/2)\left\{\exp\left(-\frac{(\tau_{r+1}-\tau_s\rho)^2}{2(1-\rho^2)}\right) - \exp\left(-\frac{(\tau_r-\tau_s\rho)^2}{2(1-\rho^2)}\right)\right\}\right\} \qquad (10)$$

for some intervals $(\tau_r,\tau_{r+1}),(\tau_s,\tau_{s+1})$ depending on $\rho$. Inspection of the above expression shows that if $|r-s|>1$ all exponentials are of the order

$$O\left(\exp\left(-\frac{c}{1-\rho^2}\right)\right) = o(\sqrt{1-\rho}) \quad \text{as } \rho\to 1, \text{ for some } c>0.$$

The case $r=s$ corresponds to a diagonal cell which do not determine the leading order. The case of a cell directly adjacent to cell (i.e. $|r-s|=1$) is crucial here. In that case we either have $\tau_{r+1}=\tau_s$ or $\tau_r = \tau_{s+1}$ so that exactly one of the four exponentials in (10) is $\Theta(1)$ as $\rho\to 1$, noting that $(1-\rho)^2/(1-\rho^2) = (1-\rho)/(1+\rho) = o(1)$ as $\rho\to 1$. In light of (9), we conclude that after having considered all summands

$$\sum_{\ell=1}^{L}\frac{\gamma_\ell(\rho)^2}{\pi_\ell(\rho)} = \Theta(1/\sqrt{1-\rho}) \text{ as } \rho\to 1.$$

Combining this with (8), we conclude the proof.

# D   Proof of Theorem 2

Let us recall some definitions from §3 of the paper. The map $\theta : (0,1) \to \mathbb{R}$ is defined by $\rho \mapsto \theta(\rho; w) = \langle \pi(\rho), w \rangle$ for $w \in \mathbb{R}^L$ such that $\langle \dot{\pi}(\rho_*), w \rangle \neq 0$. We note that by the inverse function theorem, $\theta^{-1}$ exists in a neighborhood of $\theta(\rho_*)$, and the derivative $(\theta^{-1})'$ of its inverse is given by

$$(\theta^{-1})'(\theta(\rho_*)) = \frac{1}{\dot{\theta}(\rho_*)} = \frac{1}{\langle \dot{\pi}(\rho_*), w \rangle}. \tag{11}$$

The variance of $\widehat{\theta}_w = \langle \widehat{\pi}, w \rangle$, where $\widehat{\pi}$ is the vector of the relative frequencies corresponding to $\pi = (\pi_1, \ldots, \pi_L)^\top$, is given by

$$\mathbf{Var}(\widehat{\theta}_w) = w^\top \operatorname{Cov}(\widehat{\pi})w = w^\top \Sigma(\rho_*)w/k, \tag{12}$$

where

$$\Sigma(\rho_*) := \Pi(\rho_*) - \pi(\rho_*)\pi(\rho_*)^\top,$$

and $\Pi(\rho_*) := \operatorname{diag}((\pi_\ell(\rho_*))_{\ell=1}^L)$. Here, we have used that $k\widehat{\pi}$ follows a multinomial distribution with $k$ trials and probabilities $\pi(\rho_*)$. By the law of large numbers and the continuous mapping theorem, $\widehat{\theta}_w \to \theta(\rho_*)$ in probability as $k \to \infty$. Since $\theta^{-1}$ exists in a neighborhood of $\theta(\rho_*)$, $\widehat{\rho}_w = \theta^{-1}(\widehat{\theta}_w)$ is asymptotically well-defined. By the Delta method (cf. [2], §2) and in view of (11) and (12), we have

$$\mathbf{Var}(\widehat{\rho}_w) = \frac{1}{k} \underbrace{\frac{w^\top \Sigma(\rho_*)w}{(\dot{\pi}(\rho_*)^\top w)^2}}_{V(w;\rho_*)} + O(1/k^2) \text{ as } k \to \infty.$$

which concludes the proof of the first part of the Theorem. We now turn to the proof of the second part ('Moreover, ...').

We turn to the minimization of $V(w; \rho_*)$ w.r.t. $w$ in order to determine the choice of $w$ yielding minimum asymptotic variance. Note that:

(a) $V(\alpha w) = V(w)$ for all $\alpha \in \mathbb{R} \setminus \{0\}$.

(b) $V(w + \beta \mathbf{1}) = V(w)$ for all $\beta \in \mathbb{R}$: for the numerator, this follows since $\mathbf{1}$ is contained in the null space of $\Sigma(\rho_*)$ (use that $\pi(\rho)^\top \mathbf{1} = 1$ for all $\rho$). Likewise, for the denominator $\dot{\pi}(\rho)^\top \mathbf{1} = 0$ for all $\rho$.

To eliminate these invariances, we fix the denominator and impose the constraint that $w$ be centered:

$$\min_w V_0(w), \quad V_0(w) = w^\top \Sigma(\rho_*)w, \quad \text{subject to } w^\top \mathbf{1} = 0, \ \dot{\pi}(\rho_*)^\top w = 1. \tag{13}$$

The second part of Theorem 2 can be deduced from the following lemma characterizing the preceding optimization problem.

**Lemma D.1.** *The optimal value of the optimization problem* (13) *is given by* $V_0^* = \{\dot{\pi}(\rho_*)^\top \Pi^{-1}(\rho_*)\dot{\pi}(\rho_*)\}^{-1} = I^{-1}(\rho_*)$. *The unique minimizer is given by*

$$w_0^* = V_0^* \left( \mathbf{I} - \frac{\mathbf{1}\mathbf{1}^\top}{L} \right) \Pi^{-1}(\rho_*)\dot{\pi}(\rho_*),$$

*where here* $\mathbf{I}$ *denotes the identity matrix.*

*Proof.* We first verify that $w_0^*$ is feasible.

(F1) $\quad \mathbf{1}^\top w_0^* = 0 \quad$ as $\mathbf{1}^\top \left( \mathbf{I} - \dfrac{\mathbf{1}\mathbf{1}^\top}{L} \right) = 0$.

(F2) $\quad \dot{\pi}(\rho_*)^\top w_0^* = V_0^* \dot{\pi}(\rho_*)^\top \Pi^{-1}(\rho_*)\dot{\pi}(\rho_*) = 1 \quad$ as $\dot{\pi}(\rho_*)^\top \mathbf{1} = 0$.

Note that the objective is strictly convex on the feasible set in (13): $\Sigma(\rho_*)$ is positive semidefinite whose null space equals the linear span of $\mathbf{1}$ while optimization is over its orthogonal complement.

The claim of the lemma hence follows by verifying that $w_0^*$ satisfies the KKT optimality conditions of (13). Apart from (F1) and (F2), the remaining KKT optimality condition is given by

$$2\Sigma(\rho_*)w_0^* + \vartheta^*\mathbf{1} + \gamma^*\dot{\pi}(\rho_*) = 0,$$

for Lagrangian multipliers $\vartheta^*$ and $\gamma^*$. We have

$$\Sigma(\rho_*)w_0^* = V_0^*\big(\Pi(\rho_*) - \pi(\rho_*)\pi(\rho_*)^\top\big)\left(\mathbf{I} - \frac{\mathbf{1}\mathbf{1}^\top}{L}\right)\Pi^{-1}(\rho_*)\dot{\pi}(\rho_*)$$

$$= V_0^*\big(\Pi(\rho_*) - \pi(\rho_*)\pi(\rho_*)^\top\big)\Pi^{-1}(\rho_*)\dot{\pi}(\rho_*)$$

$$= V_0^*\dot{\pi}(\rho_*)$$

Choosing $\vartheta^* = 0$ and $\gamma^* = -2V_0^*$, we conclude the assertion. The fact that $V_0^* = I^{-1}(\rho_*)$ is the minimum value of the optimization problem can be verified by substituting the expression for $w_0^*$ back into the objective. $\square$

Let $w^* = \Pi^{-1}(\rho_*)\dot{\pi}(\rho_*)$ be as in the statement of Theorem 2 in the paper. We now show that $w^*$ is related to $w_0^*$ in Lemma D.1 by scaling and translation by a multiple of $\mathbf{1}$.

$$w_0^* = V_0^*\left(\mathbf{I} - \frac{\mathbf{1}\mathbf{1}^\top}{L}\right)\Pi^{-1}(\rho_*)\dot{\pi}(\rho_*)$$

$$= V_0^*w^* - \left(\frac{\mathbf{1}^\top w^*}{L}\right)\mathbf{1}.$$

Combining this with the uniqueness of the minimizer of (13) and the invariances in (a) and (b) above, we conclude the second part of Theorem 2.

We now turn to the third part of Theorem 2. In view of the second part, we already have shown that $\mathbf{Var}(\widehat{\rho}_{w^*}) = I^{-1}(\rho_*)/k$ as $k \to \infty$. It remains to show that $\mathbf{E}[\widehat{\rho}_{w^*}] - \rho_* = O(1/k)$ as $k \to \infty$. Let $\eta = \theta^{-1}$. From a Taylor expansion, we obtain that

$$\mathbf{E}[\widehat{\rho}_{w^*}] = \mathbf{E}[\theta^{-1}(\langle\widehat{\pi}, w^*\rangle)]$$

$$= \mathbf{E}\left[\eta(\langle\pi(\rho_*), w^*\rangle) + \dot{\eta}(\theta^*)(\langle\widehat{\pi} - \pi(\rho_*), w^*\rangle) + \frac{1}{2}\ddot{\eta}(\widetilde{\theta})(\langle\widehat{\pi} - \pi(\rho_*), w^*\rangle)^2\right],$$

where $\theta^* = \theta(\rho^*; w^*) = \langle\pi(\rho_*), w^*\rangle$ and $\widetilde{\theta}$ is contained in $(\theta^*, \langle\widehat{\pi}, w^*\rangle)$. Consequently,

$$\mathbf{E}\left[\eta(\langle\pi(\rho_*), w^*\rangle) + \dot{\eta}(\theta^*)(\langle\widehat{\pi} - \pi(\rho_*), w^*\rangle) + \frac{1}{2}\ddot{\eta}(\widetilde{\theta})(\langle\widehat{\pi} - \pi(\rho_*), w^*\rangle)^2\right]$$

$$= \rho_* + \mathbf{E}\left[\frac{1}{2}\ddot{\eta}(\widetilde{\theta})(\langle\widehat{\pi} - \pi(\rho_*), w^*\rangle)^2\right] = \rho_* + O(1/k) \text{ as } k \to \infty,$$

using that $\langle\widehat{\pi}, w^*\rangle$ is an unbiased estimator of $\langle\pi(\rho_*), w^*\rangle$ with variance $O(1/k)$.

## E   Invertibility of $\theta(\cdot; w)$

We here provide some more detailed comments on the invertibility of the map $\rho \mapsto \theta(\rho; w)$. We recall that $\theta$ is invertible if and only if

$$\dot{\theta}(\rho; w) = \langle\dot{\pi}(\rho), w\rangle > 0 \quad \text{or} \quad \langle\dot{\pi}(\rho), w\rangle < 0$$

uniformly in $\rho$. As $w$ can be multiplied by $-1$, we only consider the first condition in the sequel. We first note that it is always possible to find $w$ such that this condition is satisfied. This follows from the fact that for those indices $\ell$ corresponding to diagonal cells (cf. Figure 1), $\dot{\pi}_\ell(\rho)$ is positive for all $\rho$; this can be deduced from Lemma A.1 above. As a result, any choice of $w$ with positive entries for those $\ell$ and zero else satisfies the required condition.

In light of Theorem 2, we are more interested in choices of the form $w(\rho') = \Pi^{-1}(\rho')\dot{\pi}(\rho')$. In this case, the condition for invertibility becomes

$$\langle\dot{\pi}(\rho), \Pi^{-1}(\rho')\dot{\pi}(\rho')\rangle = \sum_{\ell=1}^{L}\frac{\dot{\pi}_\ell(\rho)\,\dot{\pi}_\ell(\rho')}{\pi_\ell(\rho')} > 0.$$

uniformly in $\rho$. For $\rho = \rho'$, the left hand side equals $I(\rho')$ which is strictly positive. By continuity, this also holds in a neighborhood of $\rho'$. This neighborhood is substantial in size as $\text{sign}(\dot{\pi}_\ell(\rho)) = \text{sign}(\dot{\pi}_\ell(\rho'))$ for all $\ell$ unless $\rho$ and $\rho'$ are far apart from each other. This can be made more quantitative by evaluating $\dot{\pi}(\rho; w(\rho'))$ numerically on a dense grid of values for $\rho$ resp. $\rho'$. It turns out that the above condition is satisfied except for pairs $(\rho, \rho')$ with $\rho$ small and $\rho'$ close to one.

While this implies that global invertibility of $\theta(\cdot; w(\rho'))$ does not hold for $\rho'$ close to one, this does not affect the computational approach proposed in Section 3 of the paper. In fact, if the look-up table $\{(\rho_r, \theta(\rho_r; \rho'))\}_{r=1}^R$ with $\rho'$ depending on the pilot estimator $\widehat{\rho}_0$ does not happen to be an increasing order (which can easily be checked by computing the differences $\theta(\rho_r; \rho') - \theta(\rho_{r-1}; \rho')$, $r = 2, \ldots, R$), the look-up table can be partitioned into subsets which are ordered in either directions. The search in the look-up table can then be restricted to the subset whose range of values for $\rho$ contains the pilot estimator. This is straightforward to implement, but was not even required in any of the experiments. In fact, the deviation from global invertibility appears to be so minor that a direct binary search on the complete look-up table always delivered the correct results.

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

# Part II: Additional Figures

## F   Fisher information and empirical MSE

Full set of plots for Figure 1 (R) in the paper which only shows the case $b = 3$.

Figure 2: (Re-scaled) Empirical MSEs $k \cdot (\widehat{\rho}_{\mathrm{MLE}} - \rho_*)^2$ (averages over $10^4$ i.i.d. data sets with $k = 100$) compared to the inverse information $I^{-1}(\rho)$. The disagreement for $\rho \leq 0.2$ results from positive truncation of the MLE at zero. The thresholds have been chosen according to Lloyd-Max quantization.

# G   Choice of the thresholds

Full set of plots for Figure 2 in the paper which only shows the cases $b \in \{2, 4, 6\}$.

Figure 3: $b \cdot I_b^{-1}(\rho) / \cdot I_1^{-1}(\rho)$ vs. $\rho$ for different choices of $\mathbf{t}$: Lloyd-Max and uniform quantization with saturation levels $T_{0.9}, T_{0.95}, T_{0.99}$, cf. §4.1 in the paper for a definition. The latter are better suited for high similarity. The differences become smaller as $b$ increases.

# H Trade-off between $k$ and $b$

Full set of plots for Figure 3 (L, M) in the paper which only shows the plot for the full range of $\rho$ and for the range $\rho \geq 0.9$.

Figure 4: Trade-off between $k$ and $b$ for different ranges of $\rho$: $b \cdot I_b^{-1}(\rho)$ vs. $\rho$ for $1 \leq b \leq 6$ with $\mathbf{t}$ chosen according to Lloyd-Max quantization.

# I MLE approximation

In §3 of the paper, a scheme for approximating the MLE is discussed that only requires at most two table look-ups. We here provide empirical results that support the use of this scheme in place of the exact MLE. The MSEs show close agreement.

Figure 5: (Re-scaled) Empirical MSE $k \cdot (\widehat{\rho}_{\mathrm{MLE}} - \rho_*)^2$ for (averaged over $10^4$ i.i.d. data sets with $k = 100$) compared to the approximation discussed in §3 of the paper. While there is some slight disagreement in the small $\rho$ range, the MSEs of the approximation match those of the full MLE. The thresholds have been chosen according to Lloyd-Max quantization.

# J   $K$ Nearest neighbor search

## J.1   Results for different choices of $K$

Figure 4 in the paper only shows the case $K = 10$.

Figure 6: Average fraction of $K$, $K \in \{3, 5, 10\}$, nearest neighbors retrieved vs. total # of bits ($\log_2$ scale) for $1 \leq b \leq 6$. $b = \infty$ (dashed) represents the MLE based on unquantized data, with $k$ as for $b = 6$. The oracle curve (dotted) corresponds to $b = \infty$ with maximum $k$ (i.e. ,as for $b = 1$). For the above results, Lloyd-Max quantization was used.

## J.2 Comparison to baseline methods

Full set of plots for Figure 5 in the paper which only shows the cases $b \in \{2, 4\}$.

Figure 7: Average fraction of $K = 10$ nearest neighbors retrieved vs. total # of bits ($\log_2$ scale) of our approach (MLE) relative to that based on the Hamming distance and the linear approach.

## J.3 Influence of the choice of t

Figure 8: Average fraction of $K = 10$ nearest neighbors retrieved vs. total # of bits ($\log_2$ scale) for $1 \leq b \leq 6$. $b = \infty$ (dashed) represents the MLE based on unquantized data, with $k$ as for $b = 6$. The oracle curve (dotted) corresponds to $b = \infty$ with maximum $k$ (i.e. ,as for $b = 1$). The figure depicts the dependency on the choice of the thresholds: Lloyd-Max, $T_{0.9}$, $T_{0.95}$ and $T_{0.99}$.

# K  Linear classification

These are magnified versions of the left and middle plot in Figure 6 of the paper.

Figure 9: Accuracy vs. bits, optimized over the SVM parameter $C$.

Full set of plots for Figure 6 (R) in the paper which only shows the case in which #bits = $2^{10}$.

Figure 10: Accuracy vs. the SVM parameter $C$ for fixed budgets of bits.

Plots as in the figure above, with the farm instead of the arcene data set.

Figure 11: Acccuracy vs. the SVM parameter $C$ for fixed budgets of bits.