[Reviews · NeurIPS 2016]

Reviewer 1

Summary

The paper considers the problem of estimating cosine similarity from quantized random projections. They propose a maximum likelihood estimator and demonstrate how its better than the empirical approach. They also demonstrate the trade-off between accuracy and quantization.

Qualitative Assessment

The paper has two new contributions: extending the 1 bit quantization (LSH type) to b bits quantization and using MLE estimator instead of empirical estimator. In communications theory, there has been a lot of study on Quadrature Amplitude Modulation (QAM) which essentially studies how to choose quantizers for various types of noise (for example Gaussian in your case), they study the optimal quantizers, effect of b ( the number of bits in approximation) etc.. See for example: Quadrature Amplitude Modulation (QAM) in https://web.stanford.edu/group/cioffi/doc/book/chap1.pdf I was wondering if there any interesting connections between the two problems? Is it possible to quantize the bias for empirical estimation and MLE as a function of the bias b?

Confidence in this Review

2-Confident (read it all; understood it all reasonably well)


Reviewer 2

Summary

The paper studies random projection-based estimation of cosine similarity. It uses scalar quantization to quantize each dimension of the random projections to b bits. 0/1 quantization corresponds to the well-known simHash; the non-quantized version is is the classic result of JL lemma. The work therefore bridges the gap of the above.

Qualitative Assessment

This paper is technically strong based on the following 1. The MLE framework makes it possible to generalize the 1-bit quantization to the b-bit case, and it is shown that the 2-bit hamming distance approach in [17] is a special case of this work. 2. I particularly like the discussions on under what conditions the d-bit quantization is helpful and the trade off between k and b. The fact that phi->1 requires large b makes this work useful in real-world applications on high-dimensional data, where due the curse of dimensionality all the data points are close. This also leads to theorem 1, which sheds lights on why a JL-style bound is not generally possible under the discrete setting. The execution of the paper needs improvements based on the following. 1. There are a few technical difficulties in applying the MLE-based estimator and determining the best quantization intervals. For example, as mentioned in page 5, it requires optimization via an iterative scheme, and the estimator of \hat{\phi_w*} is not operational. I highly appreciate the efforts in providing remedies addressing such issues. However it should be made more clear what the additional cost is when applying the b-bit MLE approach. For example, the construction and storage of the lookup table take additional time and space, and the estimation of the empirical cell frequencies may not be accurate. 2. In the figures of all the experiment, I highly suggest adding two types of curves. a. The naive real-valued random projection with each dimension represented by, say, 4/8/16 bit integer and half/single/double precision float. b. A set of oracle curves (performance upper bound) assuming that the full precision random projection can be quantized by 1/2/3/4... bits without loss of precision. 3. It is worthwhile to conduct an addition experiment on some synthetic data where rho (on average) is small.

Confidence in this Review

2-Confident (read it all; understood it all reasonably well)


Reviewer 3

Summary

Analysis of the random projections algorithm for estimating cosine similarity. The authors show that MLE is the right approach to deal with quantization nonlinearity. An analysis of the trade-off between depth and the number of projections is given.

Qualitative Assessment

Nice theoretical analysis of the random projection trick for cosine similarity yet with limited practical applications. Interesting theoretical results with well written proofs. To the best of my knowledge, an interesting contribution to the overall discussion of the use of random projections in the nonlinear settings.

Confidence in this Review

3-Expert (read the paper in detail, know the area, quite certain of my opinion)


Reviewer 4

Summary

The paper studies the usage of quantized random projection for estimating inner products of unit-norm (or fixed-norm) vectors. As the problem boils down to estimating the correlation coefficient between two Gaussian random variable after quantization, the approach is based on the MLE of the correlation using the quantized projection. The paper investigates the properties of the MLE, the choice of quantizers, the tradeoff between number of bits and number of projections, and the rate of growth of the Fisher information for the high similarity regime. The performance of the procedure is evaluated on simulated dataset and real datasets in the context of nearest neighbor search and linear classification, with comparison to other existing methods.

Qualitative Assessment

The paper includes a lot of results; as a result, it seems a little bit disorganized, especially Chapter 2. It reads like an aggregation of independent results and conclusions, without a main theme that connects them all. - For starters, it'd be nice to clarify that estimating the inner product between x and x' with random projections boils down to estimating the correlation coefficient between corresponding z and z', and justify the discussion solely on z. - Secondly, I'd appreciate it if the authors could make some sort of a conclusion after listing a few options for a procedure. such as suggesting a "default" or "adaptive" algorithm that can be applied directly and having performance guarantee. - Moreover, some results seem to be a bit too empirical and qualitative. For example, the trade-off between k and b is a bit too general and seems to depend on the choice of the quantizer. For quantization, is there a reason for only considering integer values of b, as well as b no larger than 6? In addition to scalar quantization, is it possible to utilize vector quantization and improve the performance? The approximate MLE computation seems neat, but why isn't it implemented in the experiments? It would be interesting to see how much is lost due to the approximation. Regarding the experiments, for the synthetic data, the notation is a bit confusing--is X here playing the role of Z in the previous derivation? For the real data, a comment or justification on normalizing each instance is appreciated. As for the results, why is b=5 outperforms b=infty for the rcv1, K=10 case? Also, it'd be interesting to see how the algorithm works for the low similarity cases, especially when the quantizer is misspecified. Overall, the paper investigate an interesting problem, but the conclusions are a bit too qualitative and need to be polished.

Confidence in this Review

2-Confident (read it all; understood it all reasonably well)


Reviewer 5

Summary

This paper considers the problem of dimensionality reduction via random projections in the setting where the projected data are quantized to adhere to an overall bit-budget. In this setting there is an inherent tradeoff between the dimension of the space we are projecting into and the number of bits per dimension that we have available. The authors approach this problem from the perspective of preserving the inner products between pairs of unit-norm vectors and show that the optimal point on this tradeoff depends on how closely aligned the vectors can be. In particular, the authors show that when the vectors are all nearly orthogonal, the best performance is obtained by setting the dimension as high as possible, reducing the quantization to simple 1-bit quantization of each observation. On the other hand, when the vectors are very closely aligned, it is better to assign as many bits as possible to each observation. This makes intuitive sense (after a bit of thinking) since when two vectors are very close together they would agree in their first significant bit on almost every possible measurement, so more bits is probably much more important in this setting. However, this was not obvious to me a priori and I thought the paper did a good job of fleshing out this issue and exploring it with some actual datasets.

Qualitative Assessment

I actually liked the paper quite a bit, but I do have at least a few concerns. First of all, it is very important in establishing the result that the approach to estimating the inner product between the observations is *not* simply taking the inner product between the observations, but by computing the MLE of the inner product. I see how the authors do this in the Gaussian case, but for this to be relevant in practice (on truly high-dimensional data) it seems that it would be important for this to be possible with other kinds of randomized embeddings (such as the results of the Fast Johnson-Lindenstrauss Transform). Some discussion about whether or not the techniques presented would be relevant in such a setting would be welcome. My other main concern is the manner in which the authors have brushed aside the issue of normalization. If the data is arbitrarily normalized (not simply on the sphere) I think this becomes an important piece of information. The authors state that it is easy to compute these in a single scan, which I certainly agree with, but these quantities will need to be stored (and quantized!) If you are operating within a total bit-budget, it seems that the number of bits devoted to storing the norm could also be a potential factor. Perhaps the authors are assuming that this is negligible compared to the total number of bits available, but I would at least like to see some discussion of this. I would also have liked to see some discussion about what the observed results have to say about how one should generally set the number of bits. The message in many areas (e.g., Locality-sensitive hashing) is that it is more important to preserve the distances between close points and less important to preserve global distances. Does this suggest that in general 1-bit quantization will not do well in most applications? How does the normalization play into this? (Two vectors which are far apart can become extremely close once normalized to the unit-sphere.) I would have liked to see a bit more on this in the paper.

Confidence in this Review

3-Expert (read the paper in detail, know the area, quite certain of my opinion)


Reviewer 6

Summary

NIPS 2016 Review of Paper ID 1404 “Quantized Random Projections and Non-Linear Estimation of Cosine Similarity” This manuscript studies the problem of estimating cosine similarities when the projections are quantized to an arbitrary number of bits. Previous work has examined random projections with full precision (i.e. real-valued projections) and random projections that have been quantized to a single bit (i.e. binary-valued responses). This paper focuses on the middle ground. In particular, they examine a maximum likelihood estimator- presumably of cosine similarity under the quantization non-linearity--the trade-off between bit depth vs. the number of projections for storage or transmission (i.e. compression).

Qualitative Assessment

The topic is of potential interest, but the authors wrote this manuscript with a specialized audience in mind. I am a non-specialist. As a consequence, I am not in a good position to evaluate the merits or the importance of the work to the authors’ academic community. However, I am in a position to evaluate the writing and the presentation of the submitted manuscript. It is clear that the manuscript would benefit from more thorough and methodical writing. Figures and figure captions are not carefully explained; it is often not clear whether the data presented in the figures buttress the claims and conclusions made by the authors. Sometimes, parameters are used that are never defined. In general, while the statement of the problem is clear and the mathematical development is largely clear and easy to follow, the numerical results presented by the authors are very difficult to make sense of. Line 61: “for its first resp. second derivative.” What is the first resp second derivative? Typo? Line 129: “resp.” resp.? Line 131: “The assumption that the norms are known is reasonable since 1) information about the norms is lost because of quantization (a total loss occurs for b = 1 ) and 2) it is not hard to satisfy in practice as they can be computed by one linear scan during data collection.” Confusing, possibly incoherent sentence. How can it be reasonable to assume that the norms are known if information about the norms is lost. Figure 3 Caption: What do the number in the first subplot’s legend indicate? Line 268; “We perform 100 resp. 20 independent replications for synthetic resp. real data” What does ‘resp.’ mean? The authors use it repeatedly in the manuscript. I do not know what it means and I could not decipher either from context or a google search what the authors intend it to mean. Line 279: “Some of these findings are reflected in Figures 4 and 5. We refer to the supplement for additional figures.” Which findings? I could not figure out what the figures showed. The authors assert that their results can be summarized by the fact that many bit quantization is superior to one bit quantization. This seems like a sensible conclusion, but it is not clear how Figure 4 and 5 show that. Consider figure 5. First, the y-axis has a different scale for each subplot, so comparisons between b=2 and b=4 are not easy in the farm or rcv1 datasets. This is non-standard and should be rectified. Second, if one does take the time to compare the numbers for the MLE estimator (black curve), the numbers are nearly identical for both datasets. That is, it is not at all clear from the figure that a larger number of bits (2 vs 4) indeed yields better retrieval of the K nearest neighbors. It would also be nice to know whether the 0.92 fraction retrieved in the farm dataset is considered ‘good performance’. Line 290: “For SVM classification, we consider logarithmically spaced grids between 10^3 and 10^3 for the parameter C” The SVM parameter C is never described or explained. Line 301: “there is empirical evidence that using three or a higher number potentially matches the performance achieved in the full precision case.” Three or a higher number of what? Bits? Appendix Figure 1: “(Re-scaled) Empirical MSE k _ (b_MLE􀀀__ )2 for (averaged over 104 i.i.d. data sets with k = 100) compared to the inverse information.” Nonsense first sentence of caption. Appendix Figure 5: “ verage fraction of K = 10 nearest neighbors retrieved vs. total # of bits (log2 scale) of our approach (MLE) relative to that based on the Hamming distance and the linear approach.” Typos.

Confidence in this Review

1-Less confident (might not have understood significant parts)